# Corneal biomechanical predictors of intraocular pressure elevation after intravitreal anti-VEGF injection

Sayaka Sumi[1], Ryo Asaoka[2], Shuichiro Aoki[1], Kohdai Kitamoto[1], Ryo Terao[1], Mariko Kawata[1], Tatsuya Inoue[1,3], Ryo Obata[1,4], Keiko Azuma[1]*

**1** Department of Ophthalmology, Graduate School of Medicine and Faculty of Medicine, The University of Tokyo, Tokyo, Japan, **2** Department of Ophthalmology, Seirei Christopher University & Seirei Hamamatsu General Hospital, Shizuoka, Japan, **3** Department of Ophthalmology and Micro-Technology, Yokohama City University School of Medicine, Kanagawa, Japan, **4** Department of Ophthalmology and Toranomon Hospital, Tokyo, Japan

* keikoazuma719@gmail.com

## Abstract

### Purpose

To investigate whether corneal biomechanical parameters measured via Corvis ST can predict acute intraocular pressure (IOP) elevation following intravitreal anti-VEGF injection.

### Design

Retrospective observational study.

### Subjects

Forty eyes from patients with neovascular age-related macular degeneration or retinal vein occlusion who underwent anti-VEGF therapy.

### Methods

IOP was measured using the Corvis ST immediately before and 10 minutes after injection. The following biomechanical parameters were evaluated: DA Ratio MAX (2mm), biomechanically corrected IOP (bIOP), Peak Distance, Deflection Amplitude Max, Integrated Radius, and Stress-Strain Index (SSI).

### Main outcome measures

Acute post-injection IOP elevation (continuous) and IOP spikes ≥10 mmHg (binary).

### Results

The mean IOP increased significantly from 14.5±3.17 to 24.7±7.44 mmHg post-injection (p < 0.0001). IOP spikes ≥10 mmHg occurred in 55% of eyes. On

**Data availability statement:** All relevant data are within the manuscript and its Supporting Information files.

**Funding:** This study was done under JSPS KAKENHI grant number JP21K16893. The organization had no role in study design, in the collection, analysis, and interpretation of data, in the writing of the report, or in the decision to submit the article for publication.

**Competing interests:** The authors have declared that no competing interests exist.

multivariate analysis, higher bIOP ($\beta$ = +1.17, p = 0.048) and lower DA Ratio MAX ($\beta$ = −5.40, p = 0.038) were independent predictors of IOP elevation. DA Ratio MAX was the only significant predictor of IOP spikes (OR = 0.70, 95% CI: 0.51–0.96, p = 0.035). ROC analysis showed that DA Ratio MAX alone (AUC = 0.739) outperformed bIOP (AUC = 0.607), with the combined model yielding the highest AUC (0.773). A cutoff of DA Ratio MAX ≤4.936 provided 81.8% sensitivity and 42.9% specificity for predicting spikes.

## Conclusions

DA Ratio MAX (2mm), reflecting global ocular compliance, was a significant predictor of acute IOP spikes after anti-VEGF injection. Alongside bIOP, it may be useful for pre-injection risk stratification of pressure-related complications.

## Introduction

The intravitreal injection (IVT) of anti-vascular endothelial growth factor (anti-VEGF) therapy has become the standard care for retinal vascular diseases such as neovascular age-related macular degeneration (nAMD), diabetic macular edema, and retinal vein occlusion (RVO) [1]. Despite its well-established efficacy and safety, acute post-injection elevations in intraocular pressure (IOP) remain a frequently encountered complication [2,3]. It is not uncommon for post-injection IOP to transiently exceed 50–60 mmHg, and this can occasionally reach levels that compromise ocular perfusion [4]. Although most IOP spikes are transient, repeated acute elevations can negatively impact optic nerve function, particularly alongside preexisting glaucoma, where autoregulation is impaired [5,6]. Experimental studies have demonstrated that acute IOP elevations can disrupt axoplasmic flow [7] and impair retinal ganglion cell survival [8]. In addition, altered aqueous humor dynamics during repeated spikes may contribute to long-term trabecular meshwork damage [9]. Furthermore, repeated IOP spikes can also result in long-term structural damage to the trabecular meshwork, potentially leading to sustained ocular hypertension [10,11]. Thus, identifying eyes at risk for post-injection IOP elevation is of great clinical importance.

Several anatomical factors have been reported as risk factors for IOP elevation after IVT, including short axial length [11,12], phakic lens status [11], glaucoma history [5], and injection volume [13]. However, because these parameters alone do not fully explain the considerable variability observed among individuals, there has been a recent focus on evaluating ocular biomechanical properties, which may influence the ability of the eye to buffer rapid volume changes during IVT [2,14].

Noncontact devices such as the Ocular Response Analyzer (ORA; Reichert Technologies, USA) and the Corneal Visualization Scheimpflug Technology (Corvis ST; Oculus, Germany) enable the *in vivo* assessment of biomechanical behavior. ORA-derived Corneal Hysteresis has also been associated with susceptibility to IOP elevation after IVT [14], suggesting the involvement of ocular compliance in pressure regulation. However, despite the growing use of the Corvis ST, which captures

dynamic corneal deformation responses and computes a range of biomechanical parameters, no previous study has systematically evaluated the relationship between Corvis ST-derived parameters and acute IOP spikes after IVT. The Corvis ST-derived parameters reflect corneal stiffness and viscoelasticity, and possibly global ocular rigidity under certain conditions. Thus, their potential role in modulating postinjection IOP changes warrants investigation.

This study investigated the association between corneal biomechanical parameters (measured by Corvis ST) and acute IOP elevation after anti-VEGF injection, aiming to clarify whether certain biomechanical properties could serve as predictive markers for eyes with a higher risk of significant IOP elevations.

## Methods

### Study design and participants

This retrospective study initially identified 43 consecutive eyes that underwent intravitreal anti-VEGF injection at the Department of Ophthalmology, University of Tokyo Hospital, between October and December 2024. Patients diagnosed with nAMD or RVO and with complete ocular biometric data were included. Those with a history of glaucoma, previous intraocular surgery (other than uncomplicated cataract surgery), corneal disorders, and other retinal diseases were excluded. In detail, eyes with a history of primary open-angle glaucoma (n = 2) or ocular hypertension (n = 1) were excluded to avoid potential confounding of the relationship between pre-existing IOP regulation and post-injection IOP elevation. No eyes with clinically significant corneal disorders were identified. After these exclusions, 40 eyes were included in the final analysis. There were no missing data for any variables analyzed. This retrospective study was approved by the Institutional Review Board of the University of Tokyo (Ethics Committee ID number: 2217), and the requirement for informed consent was waived in accordance with institutional guidelines and relevant legislation.

### Intravitreal injection procedure

All injections were performed under topical anesthesia using a 34-gauge needle. The anti-VEGF agents administered included aflibercept (2 or 8 mg), faricimab, ranibizumab, or brolucizumab, according to the clinical indications. The injected volume was 0.05 mL for aflibercept (2 mg), ranibizumab, and brolucizumab, and 0.07 mL for aflibercept (8 mg) and faricimab.

### Axial length

Axial length was measured using IOL Master (Tomey OA-2000, version 5.4.4.0006; Tomey, Nagoya, Japan) and assessed as a potential covariate; however, its variation was minimal within this cohort (mean 24.3 ± 1.29 mm) and did not meaningfully impact the final models.

### IOP measurement and definition of IOP spike

The corneal biomechanical parameters were simultaneously recorded prior to injection. IOP was measured using the noncontact tonometer function of the Corvis ST (Oculus, Wetzlar, Germany) immediately before and 10 minutes after injection.

An intraocular pressure (IOP) spike was defined as an increase of ≥10 mmHg from the baseline IOP measured immediately prior to intravitreal injection. This threshold was selected based on prior literature and its clinical relevance in assessing transient IOP elevations following anti-VEGF therapy.

### Corneal biomechanical assessment

The principles of the Corvis ST measurements are described previously [15]. The device is equipped with a high-speed Scheimpflug camera that captures 140 images over 30 milliseconds, documenting the corneal deformation induced by an air

jet. Dynamic measurements such as applanation times, velocities, and deformation amplitudes are recorded. The analysis focuses on two applanation events, initial inward flattening (A1) and subsequent outward flattening (A2), as well as the point of maximum concavity (HC). The Corvis ST software calculates multiple biomechanical parameters based on these events (version 1.6r2223).

Due to the large number of available parameters and their intercorrelations, this study focused on the following key parameters related to ocular biomechanics, as described previously [16–18]:

- Peak Distance (mm): The horizontal distance between the two bending peaks at the point of highest concavity.

- Deflection Amplitude Max (mm): The maximum depth of corneal apex displacement relative to baseline.

- Integrated Inverse Radius (mm$^{-1}$): The integral of the inverse curvature radius across the concave deformation phase. Larger values indicate a softer, more deformable cornea.

- DA Ratio Max (2 mm): The ratio of the central deformation amplitude (within 2 mm from the apex) to the peripheral deformation amplitude at the maximum concavity. Higher values indicate relatively greater central deformability compared to the periphery, proposed as an indicator of ocular rigidity.

- Stress-Strain Index (SSI): A stiffness parameter derived from a numerical simulation model that estimates the nonlinear stress-strain behavior of the cornea. SSI reflects the intrinsic corneal stiffness independent of IOP and corneal geometry. A value of 1 corresponds to the average stiffness of a healthy 50-year-old, while higher values indicate increased material stiffness and reduced deformability.

- Stiffness Parameter at first applanation (SP-A1): Represents the dynamic resistance of the cornea during initial flattening induced by an air jet. Unlike SSI, SP-A1 is influenced by IOP and geometric factors, thereby reflecting the overall stiffness *in vivo* under physiological conditions. The combined analysis of SSI and SP-A1 enables the differentiation between intrinsic material stiffness and functional stiffness during mechanical loading.

Illustrative schematics of the peak distance, Deflection Amplitude Max, Integrated Inverse Radius, DA Ratio Max, and SP-A1 are provided using cross-sectional images of the cornea at the point of maximum concavity (HC) (Fig 1).

## Statistical analysis

Continuous variables are presented as the mean ± standard deviation. Changes in IOP were assessed using paired t-tests. Univariate linear regression analyses were performed to identify factors associated with post-injection IOP. Variables with $p < 0.1$ in the univariate analysis were entered into the multivariate linear regression models. The factors associated with clinically significant IOP elevation (ΔIOP ≥ 10 mmHg) were evaluated via logistic regression analysis. The model fit was assessed using $R^2$ and the corrected Akaike Information Criterion (AICc). All analyses were performed using JMP Pro version 17.0 (SAS Institute Inc., Cary, NC, USA), with $p < 0.05$ indicating statistical significance.

The discriminative ability of the corneal biomechanical parameters were further evaluated via receiver operating characteristic (ROC) curve analysis. The area under the ROC curve (AUC) was calculated for models using baseline biomechanically corrected IOP (bIOP) alone, DA Ratio MAX (2 mm) alone, and a combined multivariable logistic regression model incorporating both variables. The optimal cutoff value for the DA Ratio MAX was determined based on the maximum Youden Index, and the sensitivity, specificity, positive predictive value (PPV), and negative predictive value (NPV) were computed.

## Results

This study included a total of 40 eyes from 40 patients (mean age: 73.2 ± 15.2 years, 77.5% male). The diagnoses included nAMD in 35 eyes (87.5%) and RVO in 5 eyes (12.5%). The mean axial length was 24.3 ± 1.29 mm, and 40.0% of eyes were pseudophakic. The injection volumes were 0.05 mL in 33 eyes (82.5%) and 0.07 mL in 7 eyes (17.5%) (Table 1).

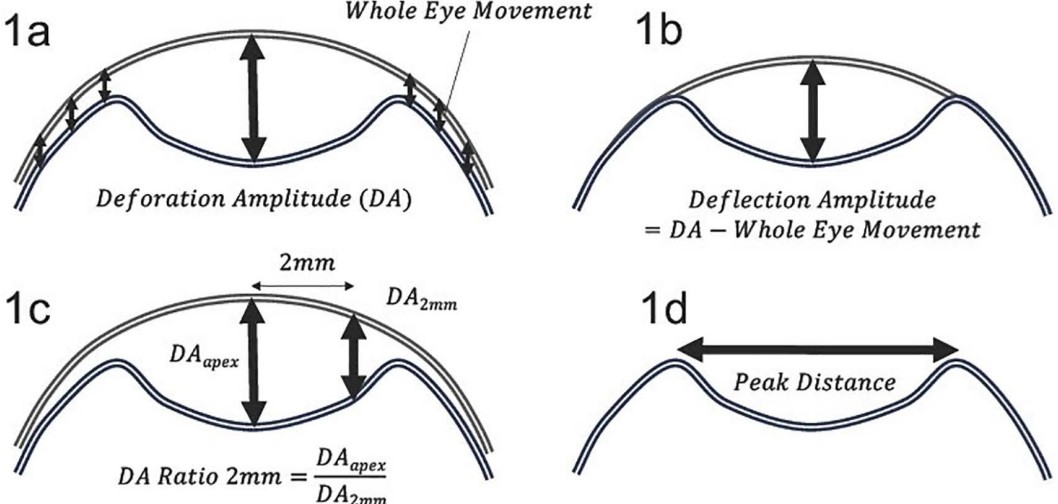

**Fig 1. Schematic representation of the Corvis ST-derived dynamic corneal response parameters.** 1a: Deformation Amplitude (DA) refers to the maximum vertical displacement of the corneal apex during indentation induced by an air jet, including both corneal and whole-eye movement components. **1b:** Deflection Amplitude is the net deformation after subtracting the whole-eye movement from the DA. **1c:** DA Ratio (2 mm) is the ratio between the apex deflection amplitude (DA_apex) and the deflection amplitude measured 2 mm from the apex (DA_2 mm); this reflects the relative deformability of the central versus peripheral cornea. **1d:** Peak Distance denotes the horizontal distance between the two corneal peaks during the highest concavity, representing the extent of spatial deformation.

**Table 1. Baseline Characteristics of Study Participants.**

| Variable | Value |
|---|---|
| Number of eyes (patients) | 40 |
| Age, years (mean ± SD) | 73.2 ± 15.2 |
| Male, n (%) | 31 (77.5%) |
| Diagnosis, n (%) | nAMD: 35 (87.5%), RVO: 5 (12.5%) |
| Axial length, mm (mean ± SD) | 24.3 ± 1.29 |
| Pseudophakic eyes, n (%) | 16 (40.0%) |
| Injection volume, n (%) | 0.07 mL: 7 (17.5%), 0.05 mL: 33 (82.5%) |

Note: SD = standard deviation; RVO = retinal vein occlusion; nAMD = neovascular age-related macular degeneration.

The mean IOP significantly increased from pre-injection to 10 minutes post-injection (14.5 ± 3.17 vs. 24.7 ± 7.44 mmHg, $p < 0.0001$, paired t-test), with an average IOP elevation of 10.2 ± 5.9 mmHg. IOP spikes of ≥ 10 mmHg and ≥ 15 mmHg were seen in 55.0% and 27.5% of eyes, respectively.

On univariate linear regression analysis, post-injection IOP was significantly associated with bIOP ($\beta = +1.31$, $p = 0.0001$), DA Ratio MAX (2 mm) ($\beta = -6.84$, $p = 0.0049$), Peak Distance ($\beta = -12.9$, $p = 0.0002$), Deflection Amplitude Max ($\beta = -37.2$, $p = 0.0003$), Integrated Radius ($\beta = -2.72$, $p = 0.017$) and SP-A1 ($\beta = +0.20$, $p = 0.0013$) (Table 2).

However, on multivariate linear regression analysis, the only variables that remained significant were bIOP ($\beta = +1.17$, $p = 0.048$) and DA Ratio MAX (2 mm) ($\beta = -5.40$, $p = 0.038$) (Table 3).

The final model demonstrated good explanatory power ($R^2 = 0.46$). The multivariate logistic regression analysis for ΔIOP ≥ 10 mmHg revealed that DA Ratio MAX (2 mm) was significantly associated with IOP spike risk (odds ratio = 0.70, 95% CI: 0.51–0.96, $p = 0.035$), indicating that a higher DA Ratio MAX reduced the risk of post-injection IOP elevation (Table 4).

**Table 2. Univariate Linear Regression Analysis for Post-injection IOP Elevation.**

| Variable | β Coefficient | p-value |
|---|---|---|
| bIOP (mmHg) | +1.31 | 0.0001 |
| DA Ratio MAX (2 mm) | −6.84 | 0.0049 |
| Peak Distance (mm) | −12.9 | 0.0002 |
| Deflection Amplitude Max (mm) | −37.2 | 0.0003 |
| Integrated Radius (1/mm) | −2.72 | 0.017 |
| Age (years) | NS | 0.59 |
| Axial Length (mm) | NS | 0.74 |
| Stress-Strain Index (SSI) | NS | 0.88 |
| SP-A1 (mmHg/mm) | +0.20 | 0.0013 |

Note: IOP = intraocular pressure; β = regression coefficient; NS = not significant; SP-A1 = stiffness parameter at first applanation; SSI = stress-strain index.

**Table 3. Multivariate Linear Regression Model for Post-injection IOP spike.**

| Variable | β Coefficient | 95% Confidence Interval | p-value |
|---|---|---|---|
| bIOP (mmHg) | +1.1685 | 0.60 to 1.74 | 0.048 |
| DA Ratio MAX (2 mm) | −5.4022 | −9.26 to −1.54 | 0.038 |

Note: IOP = intraocular pressure; β = regression coefficient; DA Ratio MAX = deformation amplitude ratio at 2 mm; bIOP = biomechanically corrected intra-ocular pressure; IOP spike = defined as an increase of ≥10 mmHg from the baseline IOP measured immediately prior to intravitreal injection.

**Table 4. Multivariate Logistic Regression Analysis for IOP Spikes ≥10 mmHg.**

| Variable | β Coefficient | Odds Ratio (OR) | 95% Confidence Interval | p-value |
|---|---|---|---|---|
| DA Ratio MAX (2 mm) | −0.40 | 0.70 | 0.51 to 0.96 | 0.035 |
| Deflection Amplitude Max (mm) | −1.24 | 0.29 | 0.07 to 1.15 | 0.086 |

Note: IOP = intraocular pressure; β = regression coefficient; OR = odds ratio; CI = confidence interval; DA Ratio MAX = deformation amplitude ratio at 2 mm.

In the ROC analysis, the multivariable model combining the DA Ratio MAX (2 mm) and bIOP showed good predictive performance (AUC = 0.773). At an optimal predicted probability cutoff (~0.56), the combined model achieved a balanced sensitivity of 72.7% and specificity of 83.3%, with a positive predictive value (PPV) of 84.2% and a negative predictive value (NPV) of 71.4%.

While bIOP alone showed limited discriminative ability (AUC = 0.607), DA Ratio MAX alone demonstrated significantly better performance (AUC = 0.739). A DA Ratio MAX cutoff of ≤ 4.936 revealed high sensitivity (81.8%) but modest specificity (42.9%), with a PPV of 56.3% and NPV of 63.6%.

## Discussion

This study investigated the relationship between corneal biomechanical parameters and acute IOP elevation after IVT anti-VEGF. Both bIOP and DA Ratio MAX (2 mm) were independently associated with post-injection IOP elevation. Notably, DA Ratio MAX (2 mm), derived from the Corvis ST device, was the only factor that consistently and significantly predicted IOP spikes (≥ 10 mmHg) in both the linear and logistic regression models.

A higher pre-injection bIOP was significantly and positively associated with post-injection IOP. Since bIOP incorporates individual corneal stiffness and geometry, it provides a more accurate reflection of the true internal pressure within

an eyeball compared to conventional tonometry methods such as Goldmann applanation tonometry or the IOPg/IOPcc values provided by the Ocular Response Analyzer [18]. These findings suggest that eyes with higher bIOP may have reduced ocular compliance, resulting in greater pressure increases in response to small volume changes from IVT [11,19]. Thus, bIOP could serve as a practical biomarker for identifying eyes with biomechanical vulnerability to pressure spikes.

Univariate analysis revealed that all Corvis ST parameters were significantly associated with post-injection IOP, including DA Ratio MAX (2 mm), Peak Distance, Deflection Amplitude MAX, Integrated Radius, and SP-A1. These parameters reflect corneal deformation induced by an air jet. As previously confirmed by Vinsinguerra et al. using dynamic Scheimpflug imaging [20], this corneal deformation is largely influenced by IOP as well, rather than just the stiffness of the cornea as a material. Thus, the Corvis ST parameters used in this study (e.g. DA Ratio MAX and Deflection Amplitude MAX) reflect ocular rigidity, which is influenced by both IOP and corneal material stiffness. These pressure-dependent deformation metrics are potentially useful indicators of acute IOP elevation following IVT [20]. On the other hand, SSI was specifically developed to represent corneal material stiffness (independent of IOP or geometric factors) using numerical simulation models that decouple the effect of IOP [18]. This may explain why the SSI was not significantly associated with IOP elevation after IVT in our analysis. Meanwhile, pressure-sensitive parameters (e.g., DA Ratio MAX) were more responsive to short-term, volume-driven IOP fluctuations, likely because they account for both material properties and IOP-induced deformation behaviors.

Among the six significant predictors identified in the univariate analysis (i.e., bIOP, DA Ratio MAX, Peak Distance, Deflection Amplitude MAX, Integrated Radius and SP-A1), the multivariate analysis revealed that only DA Ratio MAX (2 mm) remained a significant predictor of both absolute post-injection IOP and the risk of an IOP spike (≥ 10 mmHg) post-injection. Thus, DA Ratio MAX has the greatest utility as a predictor of IOP elevation after IVT, possibly because the central-to-peripheral corneal deformation profile incorporates aspects of scleral and periorbital resistance in addition to corneal properties [21–23]. In line with this, a recent study of highly myopic eyes suggested that corneal biomechanical metrics, including the DA ratio at 2 mm, may indirectly reflect scleral compliance, particularly in eyes with axial elongation [23].

In our multivariate analysis, DA Ratio MAX (2 mm) emerged as a significant predictor of acute IOP elevation, while other Corvis ST parameters such as the SSI were not. This discrepancy may reflect fundamental differences in the nature of these biomechanical metrics. SSI was developed to quantify pressure-independent material stiffness of the cornea, modeling its intrinsic stress–strain behavior regardless of IOP or corneal geometry [24,18]. In contrast, DA Ratio MAX is a dynamic deformation parameter that incorporates the corneal response under air-puff loading and is sensitive to both central and peripheral rigidity.

Recent studies have demonstrated that pressure-dependent indices, such as DA Ratio MAX, are more responsive to physiological fluctuations and may better reflect global ocular compliance under acute volume changes [25]. Notably, Dackowski et al. [14] also reported a significant association between DA Ratio and IOP elevation following intravitreal injection, supporting our findings. These results collectively suggest that dynamic deformation behavior, rather than static stiffness, may be more relevant for predicting transient IOP spikes induced by rapid intraocular volume changes. Interestingly, previous research using the Ocular Response Analyzer (ORA) demonstrated that corneal hysteresis and corneal resistance factor are related to acute IOP elevation after intravitreal injection [14]. However, their predictive performance was limited when compared to baseline IOP alone. In contrast, our findings suggest that Corvis ST-derived DA Ratio MAX may provide complementary or superior predictive value by capturing dynamic ocular compliance under real-world volume stress conditions.

Although bIOP was independently associated with absolute post-injection IOP values on multivariate analysis, it was not a significant predictor of IOP spikes (≥ 10 mmHg) in the logistic regression model. Thus, although bIOP influences IOP after IVT, this relationship is not straightforward, and its impact is better viewed in the context of ocular rigidity. In the ROC analysis, DA Ratio MAX alone demonstrated better predictive value (AUC = 0.739) compared to bIOP alone

(AUC = 0.607), while their the combined model achieved the highest discriminative ability (AUC = 0.773). Thus, DA Ratio MAX is not only an associated factor but also a robust independent predictor of acute IOP spike following IVT. The optimal cut-off value (DA Ratio MAX ≤ 4.936) achieved a sensitivity of 81.8%, suggesting its clinical utility in risk stratification prior to IVT. Building upon this conceptual distinction, we further analyzed the clinical performance of each parameter in identifying patients at risk for post-injection IOP spikes. In our ROC analysis, DA Ratio MAX demonstrated high sensitivity but modest specificity. This trade-off may limit its clinical utility as a standalone screening tool. However, when combined with baseline bIOP, the predictive performance improved markedly. These findings suggest that a multi-parametric biomechanical assessment may provide a more accurate risk stratification for transient IOP spikes following anti-VEGF injection. Collectively, these findings emphasize the complex and clinically significant interplay between intrinsic ocular pressure (i.e., bIOP), material stiffness (i.e., SSI) and pressure-dependent, geometry-sensitive deformation behavior (i.e., DA Ratio MAX). Among these factors, DA Ratio MAX integrates both local corneal properties and global ocular structural dynamics [21–23], positioning it as a promising candidate for pre-injection screening. This can be particularly useful in populations at risk of IOP-related complications, such as patients with glaucoma undergoing anti-VEGF therapy [26,27].

This study has several limitations. First, the retrospective design and relatively small sample size (40 eyes) may limit the generalizability of our findings. However, a post-hoc power analysis using the observed effect size (Cohen's d = 1.76) demonstrated that the sample size provided >99% power to detect a clinically meaningful IOP increase (ΔIOP ≥ 10 mmHg). Nevertheless, the retrospective nature of the analysis warrants cautious interpretation.

Second, although we attempted a stratified analysis by diagnosis (nAMD vs. RVO), the number of RVO eyes was limited (n = 5), making the subgroup underpowered. While no significant differences in bIOP, DA Ratio MAX (2 mm), or ΔIOP were observed between the groups, this finding remains exploratory and should be validated in larger prospective studies to evaluate disease-specific biomechanical differences. Third, IOP was only measured at a single time point (10 minutes post-injection), and thus its full temporal dynamics remain unknown. Fourth, although DA Ratio MAX showed a higher AUC than bIOP (0.739 vs. 0.607), no statistical test (e.g., DeLong's test) was performed to confirm this difference. Given the limited sample size and exploratory nature of the study, we chose to report a descriptive comparison only.

While DA Ratio MAX alone showed modest specificity, its combination with bIOP improved overall diagnostic performance. These findings support the value of using combined biomechanical metrics to better predict post-injection IOP spikes. Future studies with larger cohorts and formal AUC comparisons are warranted to confirm these results.

Furthermore, eyes with pre-existing glaucoma or ocular hypertension were excluded, and no eyes with significant corneal pathology were present in this cohort. Although axial length was assessed as a potential covariate, its limited variation did not materially affect the final models. Fifth, while lens status and injection volume may affect IOP response, an exploratory analysis including these variables showed no material change in the results, but their effects could not be fully evaluated due to the small sample size. Lastly, this study did not evaluate the effects of cumulative injections or long-term visual outcomes, which could be important in chronic treatment settings.

## Conclusion

DA Ratio MAX (2 mm) and bIOP were identified as independent predictors of acute IOP elevation following IVT anti-VEGF. DA Ratio MAX was the most robust predictor of IOP spikes ≥ 10 mmHg, positioning it as a potential biomarker of ocular compliance. Further prospective studies involving direct scleral assessment and longitudinal follow-up are needed to validate and expand on these findings.

## Glossary

- bIOP: Biomechanically corrected intraocular pressure; an intraocular pressure estimate adjusted for corneal biomechanical properties measured by Corvis ST.

- DA Ratio MAX (2 mm): Ratio of maximum deformation amplitude at the corneal apex to the average amplitude at 2 mm from the apex; reflects ocular rigidity.

- SP-A1: Stiffness parameter at the first applanation, derived from corneal deformation response; used as a marker of corneal stiffness.

- IOP spike: An acute rise in intraocular pressure ≥10 mmHg following intravitreal injection.

- Corvis ST: A non-contact tonometer that assesses dynamic corneal response using high-speed Scheimpflug imaging.

## Supporting information

**S1 File. Renamed 73ea9.** Raw data used for statistical analysis.Contains anonymized measurements for each subject and analysis grouping.
(XLSX)

**S2 Fig. renamed_c1fde.**
(TIF)

## Author contributions

**Conceptualization:** Sayaka Sumi, Keiko Azuma, Shuichiro Aoki.

**Data curation:** Keiko Azuma, Ryo Asaoka, Shuichiro Aoki, Kohdai Kitamoto, Ryo Terao, Mariko Kawata, Tatsuya Inoue, Ryo Obata.

**Formal analysis:** Keiko Azuma, Ryo Obata.

**Funding acquisition:** Sayaka Sumi, Keiko Azuma, Ryo Asaoka, Tatsuya Inoue.

**Investigation:** Keiko Azuma, Shuichiro Aoki, Kohdai Kitamoto, Mariko Kawata.

**Methodology:** Keiko Azuma, Ryo Terao, Ryo Obata.

**Project administration:** Sayaka Sumi, Keiko Azuma, Ryo Asaoka, Kohdai Kitamoto, Tatsuya Inoue.

**Resources:** Keiko Azuma, Kohdai Kitamoto.

**Software:** Keiko Azuma, Ryo Asaoka, Shuichiro Aoki.

**Supervision:** Keiko Azuma, Ryo Asaoka, Ryo Terao, Mariko Kawata, Tatsuya Inoue, Ryo Obata.

**Validation:** Sayaka Sumi, Keiko Azuma, Ryo Asaoka, Shuichiro Aoki, Tatsuya Inoue.

**Visualization:** Keiko Azuma, Ryo Terao.

**Writing – original draft:** Sayaka Sumi, Keiko Azuma.

**Writing – review & editing:** Keiko Azuma, Ryo Asaoka, Shuichiro Aoki, Kohdai Kitamoto, Ryo Terao, Tatsuya Inoue, Ryo Obata.

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
