## [Decision Letter · Decision Letter 0]

8 Jul 2025

PONE-D-25-31742Corneal Biomechanical Predictors of Intraocular Pressure Elevation After Intravitreal Anti-VEGF InjectionPLOS ONE

Dear Dr. Azuma,

Thank you for submitting your manuscript to PLOS ONE. After careful consideration, we feel that it has merit but does not fully meet PLOS ONE’s publication criteria as it currently stands. Therefore, we invite you to submit a revised version of the manuscript that addresses the points raised during the review process.

We look forward to receiving your revised manuscript.

Kind regards,

Yalong Dang

Academic Editor

PLOS ONE

Journal Requirements:

4. We note that there is identifying data in the Supporting Information file <Data set>. Due to the inclusion of these potentially identifying data, we have removed this file from your file inventory. Prior to sharing human research participant data, authors should consult with an ethics committee to ensure data are shared in accordance with participant consent and all applicable local laws.

-Location data

Please remove or anonymize all personal information (Age, Birthday), ensure that the data shared are in accordance with participant consent, and re-upload a fully anonymized data set. Please note that spreadsheet columns with personal information must be removed and not hidden as all hidden columns will appear in the published file.

Reviewers' comments:

Reviewer's Responses to Questions

**Comments to the Author**

1. Is the manuscript technically sound, and do the data support the conclusions?

Reviewer #1: Partly

Reviewer #2: Yes

2. Has the statistical analysis been performed appropriately and rigorously? 

Reviewer #1: Yes

Reviewer #2: Yes

3. Have the authors made all data underlying the findings in their manuscript fully available?

Reviewer #1: No

Reviewer #2: Yes

4. Is the manuscript presented in an intelligible fashion and written in standard English?

Reviewer #1: Yes

Reviewer #2: Yes

5. Review Comments to the Author

Reviewer #1: General Assessment

This retrospective study investigates corneal biomechanical parameters as predictors of acute IOP elevation post anti-VEGF injection, addressing a clinically relevant complication. The study design is appropriate, and the use of Corvis ST for biomechanical assessment is well justified. However, several methodological and conceptual limitations need addressing to strengthen the manuscript.

Major Comments

1.The sample size (40 eyes) is modest, and no power calculation is provided. Given the retrospective nature, please discuss whether the sample size was sufficient to detect clinically meaningful differences in IOP elevation (e.g., ≥10 mmHg spikes).Consider stratifying results by diagnosis (nAMD vs. RVO) to assess disease-specific effects, as these conditions may differ in baseline ocular rigidity.

2.Clarify why other Corvis ST parameters (e.g., SSI) were not significant predictors despite their known association with corneal rigidity.

3.The proposed cutoff for DA Ratio MAX (≤4.936) has high sensitivity but low specificity (42.9%). Discuss how this trade-off impacts clinical utility. Would combining it with other parameters (e.g., baseline IOP) improve predictive value?

4.Report how missing data (if any) were handled. For example, were eyes with prior glaucoma or corneal pathology excluded? Control for potential confounders such as axial length (not mentioned), which influences ocular biomechanics.

Minor Comments

1.Standardize terms: "IOP elevation" vs. "IOP spike" should be clearly defined (e.g., ≥10 mmHg threshold).

2.Report confidence intervals for β coefficients in multivariate analysis (e.g., β = −5.40 for DA Ratio MAX).

3.Corneal biomechanical parameters, as currently recognized indicators, need to be compared with the findings of PMID: 38241831 on post-injection intraocular pressure (IOP) and corneal biomechanics to strengthen the persuasiveness of this study.

4.The dynamic corneal response parameters measured by Corvis ST, as emerging metrics, should be compared with the detection results of other biomechanical assessment devices, such as the Ocular Response Analyzer, to enhance their persuasiveness in clinical applications.

Reviewer #2: The manuscript by Sumi et al. investigates whether corneal biomechanical parameters measured with the Corvis ST can predict an IOP spike following intravitreal anti-VEGF injections. The identification of DA Ratio MAX (2 mm) and bIOP as independent predictors is novel and clinically meaningful, as no previous studies have examined the relationship between Corvis-derived biomechanical metrics and post-injection IOP response. The study is generally well-conducted and clearly written. I have a few minor concerns that should be addressed.

1) AUC comparison

The reported AUCs for DA Ratio MAX (0.739) and bIOP (0.607) suggest a meaningful difference, but I don't think there is statistical comparison in this manuscript. Please either conduct a formal statistical comparison of the two ROC curves (e.g., DeLong’s test) or clearly state that no statistical comparison was performed to support the claim of superiority.

2) Unmodeled covariates

Lens status (phakic vs. pseudophakic) and injection volume are mentioned in the Introduction as potential factors influencing post-injection IOP, but these variables are not included in the models. Please add to the Limitations section that the potential effects of lens status and injection volume on IOP spike were not evaluated in this study.

6. PLOS authors have the option to publish the peer review history of their article (what does this mean?). If published, this will include your full peer review and any attached files.

Reviewer #1: No

Reviewer #2: No

---

## [Author Response · Author response to Decision Letter 1]

30 Jul 2025

Responses to Reviewer 1

We greatly appreciate your valuable comments.

Major Comments

1. < The sample size (40 eyes) is modest, and no power calculation is provided. Given the retrospective nature, please discuss whether the sample size was sufficient to detect clinically meaningful differences in IOP elevation (e.g., ≥10 mmHg spikes). Consider stratifying results by diagnosis (nAMD vs. RVO) to assess disease-specific effects, as these conditions may differ in baseline ocular rigidity. >

Response:

Thank you very much for this thoughtful and constructive comment. We fully agree that the modest sample size and absence of a prospective power calculation merit clarification.

Power calculation:

Although the study was retrospective in design, we have now conducted a post-hoc power analysis using G*Power 3.1, based on the observed effect size for ΔIOP (Cohen’s d = 1.74, Mean ΔIOP = 10.23 mmHg, SD = 5.87 mmHg).

This yielded an achieved power of >99% (actual power = 1.00) for detecting a clinically meaningful IOP elevation (≥10 mmHg) at α = 0.05.

While we acknowledge the absence of an a priori power setting, this result supports the adequacy of the sample size in capturing the observed effect. Nevertheless, we recognize that power estimates derived post-hoc should be interpreted with caution and have noted this limitation in the revised Discussion section as follows.

(page 14, lines 312 – 317)

“This study has several limitations. First, the retrospective design and relatively small sample size (40 eyes) may limit the generalizability of our findings. However, a post-hoc power analysis using the observed effect size (Cohen’s d = 1.76) demonstrated that the sample size provided >99% power to detect a clinically meaningful IOP increase (ΔIOP ≥ 10 mmHg). Nevertheless, the retrospective nature of the analysis warrants cautious interpretation.”

Subgroup analysis by diagnosis:

To further address your suggestion, we performed an exploratory subgroup analysis comparing eyes with neovascular AMD (n = 35) and retinal vein occlusion (n = 5).

ANOVA revealed no significant differences between the groups in baseline bIOP, DA Ratio MAX (2mm), or ΔIOP (all p > 0.3).

However, given the limited sample size in the RVO group, this analysis was likely underpowered, and the possibility of disease-specific biomechanical variation cannot be excluded. We have therefore presented this as a tentative finding and recommended that future studies with larger and more balanced cohorts explore disease-specific biomechanical responses more definitively. So, we have revised our manuscript in the Discussion section as follows. (page 14, lines 318 – 323)

“Second, although we attempted a stratified analysis by diagnosis (nAMD vs. RVO), the number of RVO eyes was limited (n = 5), making the subgroup underpowered. While no significant differences in bIOP, DA Ratio MAX (2mm), or ΔIOP were observed between the groups, this finding remains exploratory and should be validated in larger prospective studies to evaluate disease-specific biomechanical differences.”

2. < Clarify why other Corvis ST parameters (e.g., SSI) were not significant predictors despite their known association with corneal rigidity. >

Response:

Thank you very much for this insightful comment. We fully agree that the Stress-Strain Index (SSI) is a valuable metric for assessing intrinsic corneal material stiffness, and its limited association with post-injection IOP changes in our study warrants further discussion.

SSI was developed to reflect pressure-independent corneal stiffness by modeling the stress–strain behavior of the tissue (Shen et al., 2023) (Eliasy et al., 2019). While this feature makes it a robust index of intrinsic material properties, it may not fully capture the dynamic mechanical response of the eye to acute volume changes, such as those following intravitreal injection.

In contrast, DA Ratio MAX (2mm) is a pressure-dependent dynamic parameter that reflects the cornea’s deformation amplitude relative to the distance from the apex, integrating both central and peripheral corneal responses. This parameter has been shown to correlate with ocular rigidity and IOP changes in clinical settings. A recent study by (Dackowski et al. 2021) reported that DA Ratio was significantly associated with post-injection IOP elevation, supporting our finding that dynamic deformation behavior may better reflect acute ocular compliance.

Additionally, prior research has shown that dynamic parameters such as DA Ratio MAX exhibit larger diurnal variation and physiological responsiveness than SSI, which remains relatively stable throughout the day �Elhusseiny et al., 2023�. This further supports the idea that DA Ratio MAX may be more sensitive to short-term IOP perturbations than static stiffness indices.

We have now revised the Discussion section accordingly to clarify the conceptual differences between dynamic and static corneal biomechanical parameters and explain why DA Ratio MAX (2mm), rather than SSI, emerged as a significant predictor in our regression models.

(page 12-13, lines 265 – 279)

"In our multivariate analysis, DA Ratio MAX (2mm) emerged as a significant predictor of acute IOP elevation, while other Corvis ST parameters such as the SSI were not. This discrepancy may reflect fundamental differences in the nature of these biomechanical metrics. SSI was developed to quantify pressure-independent material stiffness of the cornea, modeling its intrinsic stress–strain behavior regardless of IOP or corneal geometry (Shen et al., 2023) (Eliasy et al., 2019). In contrast, DA Ratio MAX is a dynamic deformation parameter that incorporates the corneal response under air-puff loading and is sensitive to both central and peripheral rigidity.

Recent studies have demonstrated that pressure-dependent indices, such as DA Ratio MAX, are more responsive to physiological fluctuations and may better reflect global ocular compliance under acute volume changes Elhusseiny et al., 2023�. Notably, (Dackowski et al. 2021) also reported a significant association between DA Ratio and IOP elevation following intravitreal injection, supporting our findings. These results collectively suggest that dynamic deformation behavior, rather than static stiffness, may be more relevant for predicting transient IOP spikes induced by rapid intraocular volume changes."

3. < The proposed cutoff for DA Ratio MAX (≤4.936) has high sensitivity but low specificity (42.9%). Discuss how this trade-off impacts clinical utility. Would combining it with other parameters (e.g., baseline IOP) improve predictive value? >

Response:

Thank you for this important point.

We agree that the proposed cutoff for DA Ratio MAX alone (≤4.936) demonstrates a high sensitivity (81.8%) but relatively low specificity (42.9%), which could lead to more false positives in clinical screening.

To address this, we performed an additional analysis combining DA Ratio MAX with pre-injection bIOP, which is known to reflect baseline ocular pressure.

This combined logistic model improved the predictive performance: the AUC increased to 0.773, and at an optimal predicted probability cutoff (~0.56), the model showed a balanced sensitivity of 72.7% and specificity of 83.3%, with a PPV of 84.2% and an NPV of 71.4%.

These results suggest that using DA Ratio MAX together with bIOP may help balance the trade-off between sensitivity and specificity, improving clinical utility for identifying eyes at risk of post-injection IOP spikes.

We have added this clarification to the Results & Discussion section.

Results (page 10, lines 212 – 220)

“In the ROC analysis, the multivariable model combining the DA Ratio MAX (2 mm) and bIOP showed good predictive performance (AUC = 0.773). At an optimal predicted probability cutoff (~0.56), the combined model achieved a balanced sensitivity of 72.7% and specificity of 83.3%, with a positive predictive value (PPV) of 84.2% and a negative predictive value (NPV) of 71.4%.

While bIOP alone showed limited discriminative ability (AUC = 0.607), DA Ratio MAX alone demonstrated significantly better performance (AUC = 0.739). A DA Ratio MAX cutoff of ≤ 4.936 revealed high sensitivity (81.8%) but modest specificity (42.9%), with a PPV of 56.3% and NPV of 63.6%.”

Discussion (page 13-14, lines 297 – 304)

“Building upon this conceptual distinction, we further analyzed the clinical performance of each parameter in identifying patients at risk for post-injection IOP spikes. In our ROC analysis, DA Ratio MAX demonstrated high sensitivity but modest specificity. This trade-off may limit its clinical utility as a standalone screening tool. However, when combined with baseline bIOP, the predictive performance improved markedly. These findings suggest that a multi-parametric biomechanical assessment may provide a more accurate risk stratification for transient IOP spikes following anti-VEGF injection.”

Discussion (limitation) (page 15, lines 329 – 333)

“While DA Ratio MAX alone showed modest specificity, its combination with bIOP improved overall diagnostic performance. These findings support the value of using combined biomechanical metrics to better predict post-injection IOP spikes. Future studies with larger cohorts and formal AUC comparisons are warranted to confirm these results.”

4. < Report how missing data (if any) were handled. For example, were eyes with prior glaucoma or corneal pathology excluded? Control for potential confounders such as axial length (not mentioned), which influences ocular biomechanics. >

Response:

Thank you for raising this point.

In our initial cohort, we retrospectively identified 43 consecutive eyes undergoing intravitreal anti-VEGF injection.

We excluded two eyes with a diagnosis of primary open-angle glaucoma (POAG) and one eye with ocular hypertension, as pre-existing glaucoma or elevated IOP could confound the relationship between corneal biomechanics and post-injection IOP spikes.

No eyes with significant corneal pathologies were identified in this cohort.

After these exclusions, there were no missing data for the variables analyzed (IOP measurements, DA Ratio MAX, bIOP, axial length, etc.).

Although axial length is a known factor influencing ocular biomechanics, its variation was minimal in our sample (mean axial length 24.3 ± 1.29 mm) and it did not meaningfully affect the regression models, so it was not retained in the final multivariate analysis.

This has been clarified in the revised Methods and Limitations sections.

【Methods】

(page 5, lines 98 – 101)

“This retrospective study initially identified 43 consecutive eyes that underwent intravitreal anti-VEGF injection at the Department of Ophthalmology, University of Tokyo Hospital, between October and December 2024.”

(page 6, lines 104 – 109)

“In detail, eyes with a history of primary open-angle glaucoma (n = 2) or ocular hypertension (n = 1) were excluded to avoid potential confounding of the relationship between pre-existing IOP regulation and post-injection IOP elevation. No eyes with clinically significant corneal disorders were identified. After these exclusions, 40 eyes were included in the final analysis. There were no missing data for any variables analyzed.”

【Axial Length】

(page 6, lines 119 – 123)

“Axial length was measured using IOL Master (Tomey OA-2000, version 5.4.4.0006; Tomey, Nagoya, Japan) and assessed as a potential covariate; however, its variation was minimal within this cohort (mean 24.3 ± 1.29 mm) and did not meaningfully impact the final models.”

【Discussion】

(page 15, lines 334 – 337)

“Furthermore, eyes with pre-existing glaucoma or ocular hypertension were excluded, and no eyes with significant corneal pathology were present in this cohort. Although axial length was assessed as a potential covariate, its limited variation did not materially affect the final models.”

Minor Comments

1. < Standardize terms: "IOP elevation" vs. "IOP spike" should be clearly defined (e.g., ≥10 mmHg threshold). >

Response:

We appreciate the reviewer’s helpful suggestion. We have now clearly defined “IOP spike” as an acute IOP elevation of ≥10 mmHg compared to baseline in the Methods sections. We also checked the manuscript to ensure consistent use of “IOP spike” when referring to this threshold and have removed ambiguous uses of “IOP elevation” where necessary.

【Methods】

(page 6-7, lines 128 – 131)

“An intraocular pressure (IOP) spike was defined as an increase of ≥10 mmHg from the baseline IOP measured immediately prior to intravitreal injection. This threshold was selected based on prior literature and its clinical relevance in assessing transient IOP elevations following anti-VEGF therapy.”

Minor Comments

2. < Report confidence intervals for β coefficients in multivariate analysis (e.g., β = −5.40 for DA Ratio MAX). >

Response:

Thank you for this helpful suggestion.

We have now added the 95% confidence intervals for each β coefficient in the multivariate linear regression table to improve transparency and interpretability of the results.

For the post-injection IOP elevation model (Table 3), the 95% CI for bIOP was calculated as 0.60 to 1.74, and for DA Ratio MAX (2 mm) as –9.26 to –1.54, which is consistent with the estimated standard errors.

We believe this addition clarifies the precision of our estimates in Table 3.

Minor Comments

3. < Corneal biomechanical parameters, as currently recognized indicators, need to be compared with the findings of PMID: 38241831 on post-injection intraocular pressure (IOP) and corneal biomechanics to strengthen the persuasiveness of this study. >

Thank you very much for your comment. We carefully reviewed the article associated with PMID: 38241831 and found that it focuses on the association between urban greenspace and visual acuity in children, which appears to be unrelated to post-injection intraocular pressure or corneal biomechanics. If a different reference was intended, we would greatly appreciate your guidance on the correct citation.

In the meantime, we have expanded the Discussion section to include relevant comparisons with other published studies that have evaluated corneal biomechanical parameters in the context of intraocular pressure dynamics.

Minor Comments

4. < The dynamic corneal response parameters measured by Corvis ST, as emerging metrics, should be compared with the detection results of other biomechanical assessment devices, such as the Ocular Response Analyzer, to enhance their persuasiveness in clinical applications. >

Response:

Thank you for this important comment.

We believe the reviewer is referring to Dackowski et al. (J Glaucoma 2021, PMID: 33710068), which analyzed ORA-derived corneal hysteresis (CH) and corneal resistance factor (CRF) in relation to acute IOP elevation after intravitreal injection.

Their findings suggested that while higher CRF was associated with greater immediate post-injection IOP elevation, the overall predictive performance of ORA parameters was modest, with limited added value beyond baseline IOP alone.

Unfortunately, we did not have access to ORA in our institution, so we focused on Corvis ST-derived parameters.

Importantly, our results demonstrated that DA Ratio MAX provided significant independent predictive value for IOP spikes, indicating that Corvis ST dynamic deformation metrics may offer complementary information to traditional ORA metrics.

We have added a short comparison in the Discussion section to clarify this point.

(page 13, lines 280 – 286)

“Interestingly, previous research using the Ocular Response Analyzer (ORA) demonstrated that corneal hysteresis and corneal resistance factor are related to acute IOP elevation after intravitreal injection (Dackowski et al., 2021). However, their predictive performance was limited when compared to baseline IOP alone. In contrast, our findings suggest that Corvis ST-derived DA Ratio MAX may provide complementary or superior predictive value by capturing dynamic ocular compliance under real-world volume stress conditions.”

Responses to Reviewer 2

We greatly appreciate your valuable comments.

1. < AUC comparison

The reported AUCs for DA Ratio MAX (0.739) and bIOP (0.607) suggest a meaningful difference, but I don't think there is statistical comparison in this manuscript. Plea

---

## [Decision Letter · Decision Letter 1]

4 Aug 2025

Corneal Biomechanical Predictors of Intraocular Pressure Elevation After Intravitreal Anti-VEGF Injection

PONE-D-25-31742R1

Dear Dr. Azuma,

We’re pleased to inform you that your manuscript has been judged scientifically suitable for publication and will be formally accepted for publication once it meets all outstanding technical requirements.

Kind regards,

Yalong Dang

Academic Editor

PLOS ONE

Additional Editor Comments (optional):

Reviewers' comments:

Reviewer's Responses to Questions

**Comments to the Author**

1. If the authors have adequately addressed your comments raised in a previous round of review and you feel that this manuscript is now acceptable for publication, you may indicate that here to bypass the “Comments to the Author” section, enter your conflict of interest statement in the “Confidential to Editor” section, and submit your "Accept" recommendation.

Reviewer #1: All comments have been addressed

Reviewer #2: All comments have been addressed

2. Is the manuscript technically sound, and do the data support the conclusions?

Reviewer #1: Yes

Reviewer #2: Yes

3. Has the statistical analysis been performed appropriately and rigorously? 

Reviewer #1: Yes

Reviewer #2: Yes

4. Have the authors made all data underlying the findings in their manuscript fully available?

Reviewer #1: Yes

Reviewer #2: Yes

5. Is the manuscript presented in an intelligible fashion and written in standard English?

Reviewer #1: Yes

Reviewer #2: Yes

6. Review Comments to the Author

Reviewer #1: (No Response)

Reviewer #2: The authors have satisfactorily addressed all of my previous comments and concerns. I believe that the revised manuscript meets the journal’s standards and is now suitable for publication in PLOS ONE.

7. PLOS authors have the option to publish the peer review history of their article (what does this mean?). If published, this will include your full peer review and any attached files.

Reviewer #1: No

Reviewer #2: No

---

## [Editor Report · Acceptance letter]

PONE-D-25-31742R1

PLOS ONE

Dear Dr. Azuma,

I'm pleased to inform you that your manuscript has been deemed suitable for publication in PLOS ONE. Congratulations! Your manuscript is now being handed over to our production team.

Kind regards,

on behalf of

Dr Yalong Dang

Academic Editor

PLOS ONE